# Destruction of Fibroadenomas Using Photothermal Heating of Fe₃O₄ Nanoparticles: Experiments and Models

**Ivan B. Yeboah [1], Selassie Wonder King Hatekah [1], Yvonne Kafui Konku-Asase [1], Abu Yaya [2] and Kwabena Kan-Dapaah [1,\*]**

[1] Department of Biomedical Engineering, School of Engineering Sciences, University of Ghana, P.O. Box. LG 77, Legon Accra, Ghana; iv_yeb@yahoo.com (I.B.Y.); hwk.selassie@gmail.com (S.W.K.H.); yvonnekafui@gmail.com (Y.K.K.-A.)

[2] Department of Materials Science and Engineering, School of Engineering Sciences, University of Ghana, P.O. Box. LG 77, Legon Accra, Ghana; ayaya@ug.edu.gh

\* Correspondence: kkan-dapaah@ug.edu.gh; Tel.: +233-23-536-4134

**Abstract:** Conventionally, observation (yearly breast imaging) is preferred to therapy to manage small-sized fibroadenomas because they are normally benign tumors. However, recent reports of increased cancer risk coupled with patient anxiety due to fear of malignancy motivate the need for non-aggressive interventions with minimal side-effects to destroy such tumors. Here, we describe an integrated approach composed of experiments and models for photothermal therapy for fibroadenomas destruction. We characterized the optical and structural properties and quantified the heat generation performance of Fe₃O₄ nanoparticles (NPs) by experiments. On the basis of the optical and structural results, we obtained the optical absorption coefficient of the Fe₃O₄ NPs via predictions based on the Mie scattering theory and integrated it into a computational model to predict in-vivo thermal damage profiles of NP-embedded fibroadenomas located within a multi-tissue breast model and irradiated with near-infrared 810 nm laser. In a series of temperature-controlled parametric studies, we demonstrate the feasibility of NP-mediated photothermal therapy for the destruction of small fibroadenomas and the influence of tumor size on the selection of parameters such as NP concentration, treatment duration and irradiation protocols (treatment durations and laser power). The implications of the results are then discussed for the development of an integrated strategy for a noninvasive photothermal therapy for fibroadenomas.

**Keywords:** magnetite nanoparticles; Mie scattering theory; near infrared laser; photothermal therapy; finite element method; bioheat transfer; diffusion approximation; Arrhenius integral; breast cancer

## 1. Introduction

Fibroadenoma is one of the commonest benign female breast diseases. Histologically, it is a well-circumscribed homogeneous biphasic solid lump with distinct imaging features made up of epithelial and stromal tissues [1]. Definitive diagnostic techniques include ultrasound, mammography, magnetic resonance imaging or stereotactic guided needle biopsy [2]. Their sizes are normally small (<2.5 cm), but can become giant juvenile tumors (>10 cm) during puberty or pregnancy [3] causing considerable pain and cosmetic deformity of the breast. Although it accounts for 25% of all breast masses in women [4], the numbers are higher in adolescents: 68% of all breast masses and 44–94% of biopsied breast lesions [5,6]. Furthermore, available data seem to suggest that incidence and recurrence rates are common in black race [7–9], who are more likely to develop breast cancer at a younger age [10].

Management of fibroadenomas can take two forms: observation and therapy. For fibroadenomas that cause pain, deform the breast, persist without any regression and are histologically complex, therapy is warranted [2]. Available options include open surgical excision as well as several modern minimally invasive probe-based thermal therapies including cryotherapy, radiofrequency ablation (RFA), microwave ablation (MWA), focused ultrasound (FUS) and laser-induced thermotherapy (LITT) [11,12]. On the other hand, observation, which involves yearly breast imaging, is usually recommended when the tumor is asymptomatic, small and not rapidly increasing in size to cause cosmetic deformity and pain. However, there are situations when patients who qualify for observation agitate due to the fear of malignancy leading to significant anxiety [12]. Furthermore, a recent study reported a 41% increase in cancer risk for women diagnosed with fibroadenomas compared to those without them [13]. Issues related to superficial skin burns, hemorrhage and hematoma, cost and complexity of technique that are associated with options stated earlier limit their use for small-sized fibroadenoma [11,14]. An ideal treatment will be one that is noninvasive with no side-effects. Recent advances in nanomedicine offer the opportunity for the design of smart strategies that can potentially overcome drawbacks with conventional techniques to reduce invasiveness and complexity.

Nanomedicine involves the use of nanomaterials—metallic and ceramic (iron-oxide) nanoparticles (NPs)—for theranostic purposes in living organisms. Photothermal therapy (PTT) is an emerging localized cancer treatment whereby NPs embedded in the tumor convert near-infrared light, which is minimally absorbed by biological tissue, to heat leading cell death. Traditionally, metallic NPs such as gold, silver, copper as well as carbon-nanotubes or graphene have been used for PTT [15]. Although several promising results have been reported in the literature for both in-vitro (cells) and in-vivo (animals), issues related to NP biocompatibility and stability have limited their progression to the clinics [15]. Unlike their metallic counterparts, ceramic NPs—$Fe_3O_4$ and $\gamma$-$Fe_2O_3$—have been used in human trials for magnetic hyperthermia treatment of brain [16] and prostate [17] cancers. Furthermore, these ceramic NPs have very recently been tested for photothermal therapy in both in-vitro and in-vivo studies. Chu et al. [18] showed that various shapes of $Fe_3O_4$ nanoparticles (NPs) were able to kill cancer cells and tumors in in-vitro (esophageal cancer cell) and in-vivo (mouse esophageal tumor) models, respectively. In another study, Espinosa and co-workers [19], demonstrated the ability of the iron-oxide NPs to act as magnetic and photothermal agents simultaneously—so called magnetophotothermal approach—and showed their unprecedented heating powers and remarkable heating efficiencies (up to 15-fold amplifications).

Here, we describe an integrated approach composed of experiments for NP characterization and models for optical property predictions and computational treatment planning. Our long term goal is to develop a noninvasive but highly efficacious treatment method for the destruction of fibroadenomas. The feasibility of such integrated approaches for photothermal therapies have been previously reported for different application in the literature [20,21]. We characterized the material properties and quantified the photothermal heat generation of $Fe_3O_4$ NPs by experimental measurements, obtained their optical absorption coefficient via experimentally guided Mie scattering theory and integrated it into a computational—finite element method (FEM)—model to predict in-vivo thermal damage of a NP-embedded tumor located in a multi-tissue breast model during irradiation by a near-infrared (NIR) 810 nm laser. Using a temperature-controlled parametric study, we explored the feasibility of NP-mediated photothermal therapy for the destruction of fibroadenomas and the influence of tumor size on parameters such as NP concentration, treatment duration and irradiation protocols (laser power and duration). The implications of the results are discussed for the development of an integrated strategy for photothermal therapy for the destruction of fibroadenomas.

## 2. Results

**Optical and structural characterization of $Fe_3O_4$ NPs.** Structural characterization of the $Fe_3O_4$ NPs—purchased commercially—were done to verify the specification provided by the manufacturer and also predict the optical absorption coefficient. X-ray diffraction spectra of the $Fe_3O_4$ NPs

revealed the presence of peaks at $2\theta$ = 31.5°, 35.8°, 38.35°, 42.75°, 47.2°, 54.04°, 57.24°, and 62.75° (Figure 1a). The observed peaks correspond to diffraction planes: (220), (311), (222), (400), (110), (422), (511), and (440), which have been attributable to cubic spinel phase of $Fe_3O_4$ (space group, *Fd-3m*, JCPDS-#19-0629). Since no other prominent phase was detected, the result implied that the NPs are essentially crystalline $Fe_3O_4$. Transmission electron method (TEM) confirmed the morphology of the NPs to be spherical (with agglomerations) and size distribution to be between 15 and 20 nm in diameter as indicated by the manufacturer (Figure 1b). The agglomeration revealed in the TEM image have been attributed to dipolar coupling between the NPs [22,23]. For any NP, its NIR photothermal effects are controlled by their NIR optical absorbance. UV-vis-NIR spectra of the NPs showed an extended optical absorption that slowly increased in the NIR region relative to the visible light region (Figure 1c). The absorbance intensity at 810 nm increased linearly with concentration, from 0.35 ([$Fe_3O_4$] = 6 mM) to 1.51 ([$Fe_3O_4$] = 24 mM) (Figure 1d). The absorbance band in the NIR region of UV-vis-NIR optical spectra is consistent with the results in the literature and has been attributed to multiple charge (electron) transfer [24]. Furthermore, the linear increase of absorbance for the range of concentration tested in this work has been previously reported elsewhere [19,25]. Shen and co-workers [25] showed that saturation starts occurring at high concentration (100 mM, absorbance values > 3 at 808 nm). In an effort to translate the experimentally measured photothermal heat generation capabilities of the $Fe_3O_4$ NPs tested in this study, we followed the flow-chart shown in Figure 1e to obtain the extinction cross sections of the MNPs, which was then used in Equation (2) to predict absorbance, $A_{pred}$. The validity of $A_{pred}$ was tested by evaluating its agreement with the experimentally measured absorbance, $A_{exp}$, for the different concentrations of $Fe_3O_4$ (6, 12, 24 mM). We observed that the predictions agreed reasonably well with experiments to within 2% for all concentrations when the sample size, $n$, in Equation (5) was equal to 5 (see Figure 1f).

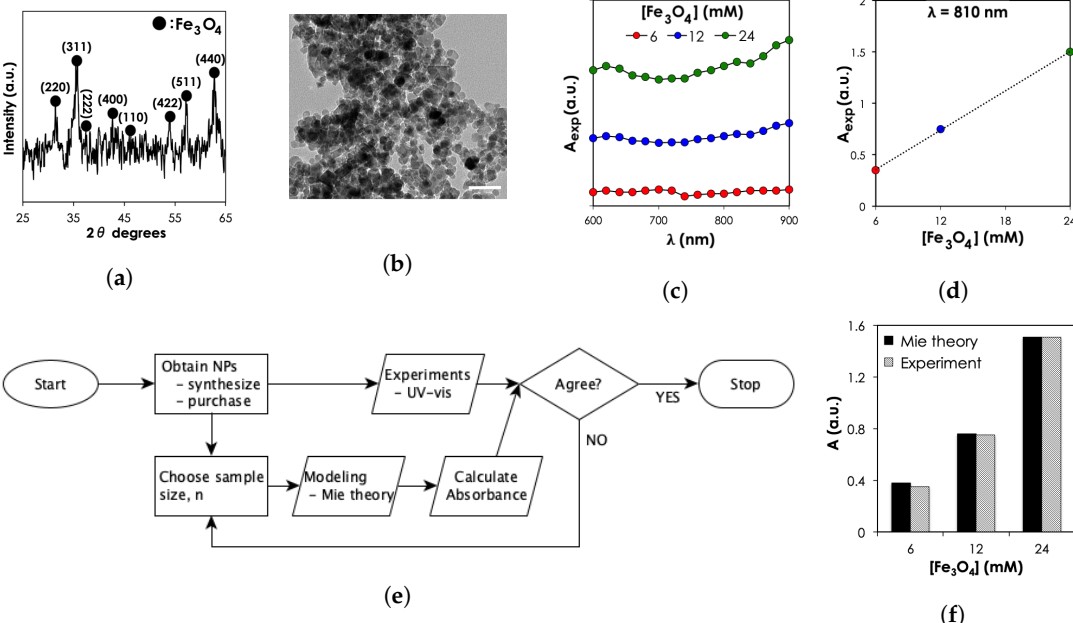

**Figure 1.** Structural and optical characterization results. (**a**) X-ray diffraction spectra at a power of 45 kV × 40 mA. (**b**) Transmission electron microscopy at magnification of 0.5 mm, Scale bar: 50 nm. Absorbance as a function of (**c**) wavelength ($\lambda$) and (**d**) concentration ([$Fe_3O_4$]) at $\lambda$ = 810 nm. (**e**) Flow-chart for the comparison of theoretical predictions and experiment measurements. (**f**) Comparison of the $A_{pred}$ and $A_{exp}$ for different [$Fe_3O_4$] (6, 12, 24 mM).

**Photothermal effects of Fe$_3$O$_4$ NPs.** The influence of laser power ($P_0$ = 0.5 and 1.0 W) and NP concentration ([$Fe_3O_4$] = 0–24 mM) on photothermal effects was accessed in aqueous solution (deionized water) to quantify their heat generation capabilities under an irradiation duration of 5 min. Pure

deionized water—containing no $Fe_3O_4$ nanoparticles—was used as a control. The rate of change of the temporal curves increased with concentration at 5 min independent of the laser power that was used (Figure 2a,b). For $P_0 = 0.5$ W, the temperature change, $\Delta T$, increased approximately by 44.4% (from $\approx 9$ to 13 °C) when concentration was increased from 0 to 24 mM (Figure 2c). When the power was increased to 1.0 W, $\Delta T$ increased by approximately 83.3% (from $\approx 12$ to 22 °C) for the same concentration. Photothermal conversion efficiency, $\eta_{exp}$, decreased with concentration and laser power (Figure 2d). For instance, $\eta_{exp}$ for the 6 mM solution decreased from approximately 66% to 51% when $P_0$ was increased from 0.5 to 1.0 W. Furthermore, when the concentration was increased from 6 to 24 mM, $\eta_{exp}$ decreased from 46% to 39% using the same power regimes. Generally, the trend of $\Delta T$ recorded in this study was in agreement with measured absorbance properties and also consistent with previously reported studies [18,19,26]. For small NPs (<30 nm) and low concentrations, absorption dominates scattering leading to high $\eta_{exp}$. On the other hand, scattering dominates the extinction efficiency as nanoparticle size or concentration is increased. As $[Fe_3O_4]$ increases, clusters are formed due to the high surface area to volume ratio of nanoparticles [27]. These clusters act as large particles to enhance scattering leading to the reduction in $\eta_{exp}$ [28]. Several approaches are available for the prevention of clusters.

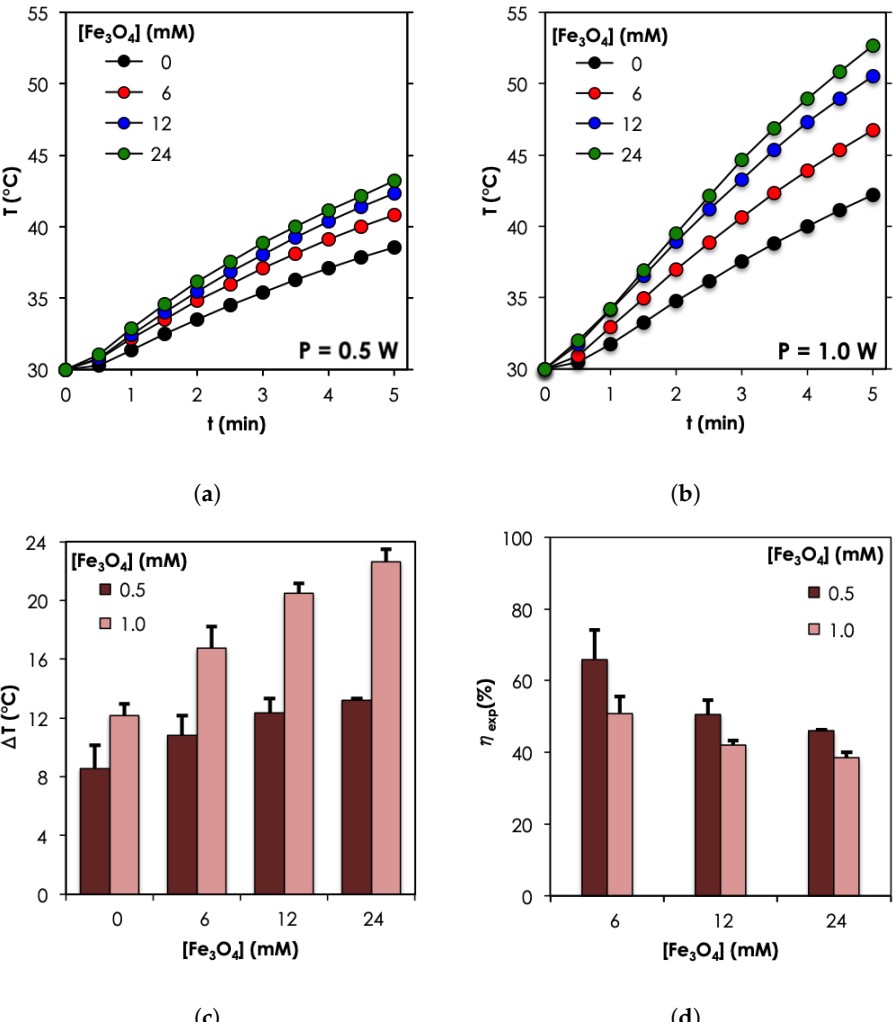

**Figure 2.** Photothermal characterization results. Temporal response curves for different concentrations after 5 min of irradiation with laser powers: (**a**) $P_0 = 0.5$ W and (**b**) $P_0 = 1.0$ W. Comparison of the corresponding (**c**) temperature change ($\Delta T$) and (**d**) experimental photothermal conversion efficiency ($\eta_{exp}$) as a function of laser power. Error bars: s.d.

**Computational modeling of NP-mediated photothermal heating of breast tumor.** The use of computational model as quantitative frameworks enables assessment and customization of the treatment parameters (NP concentration, treatment duration and irradiation protocols: duration and laser power) to potentially enhance efficacy. Thus, FEM simulations were applied to approximate photothermal heating of a $Fe_3O_4$-containing tumor embedded within a female breast using the optical diffusion approximation of the transport theory [29] and the Pennes bioheat transfer equation [30].

Figure 3 shows a schematic of 2D representation of the axisymmetric geometry of the computational model. It was configured as a heterogeneously dense [31] multi-layer block of tissue with proportions assigned according to the Breast Imaging Reporting and Data System (BIRADS) developed by American Cancer Research [32]. It consisted of various layers of normal tissue with unequal thickness. The dimensions of the model were chosen to represent a "heterogeneously dense" breast model [31], which consists of 20% muscle layer, 60% glandular layer and 20% fat layer. Also, a tumor is located at 55 mm from the base. The laser source was assumed to be a diode laser 810 nm placed close to the top surface of the breast model. The inset is a fragment of geometry showing control points P1−P4, where temperatures were recorded. The assigned optical, thermal and physical properties of different tissue layers were approximate values obtained from the literature [31,33–36]. Nanoparticles were assumed to be intravenously injected and uniformly distributed throughout the tumor.

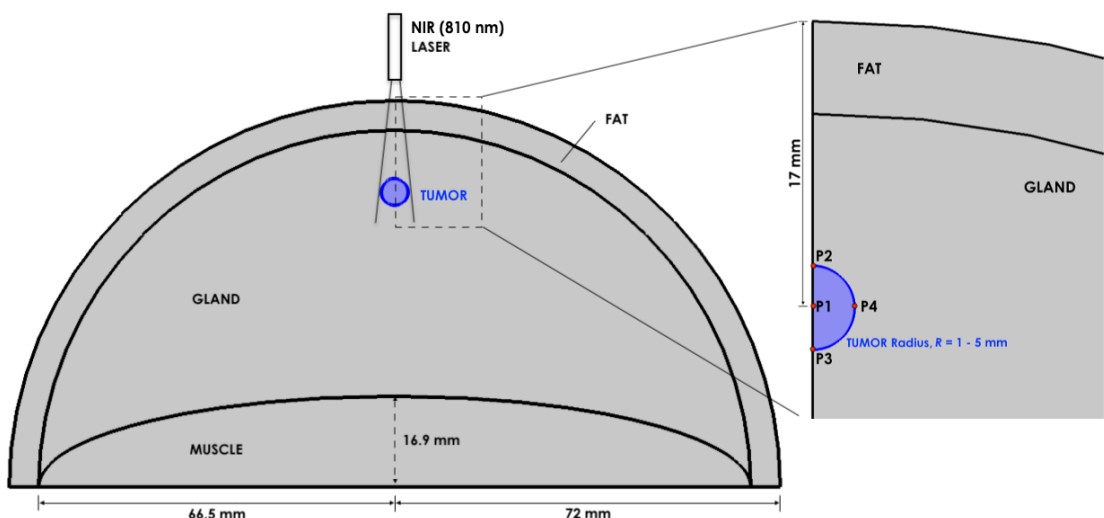

**Figure 3.** FEM geometry. Schematic of the photothermal therapy consisting of a normal multi-tissue breast domain with an embedded spherical tumor (blue sphere) and NIR (810 nm) laser source. Inset: Fragment of geometry showing controls point P1−P4, where temperature were recorded.

To characterize the temperature and thermal damage profiles, we simulated temperature-controlled heating at a maximum tumor temperature, $T_{max} = 85\ °C$, for $t = 15$ min. The radius of the tumor, $R$, and $P_0$, were chosen to be 2.5 mm and 1 W respectively. The predicted temperature distribution (Figure 4a) was revealed to be non-uniform with the maximum temperature occurring within the tumor and decreasing radially outwards into the surrounding tissue. The latter suggests that the heat transfer was predominantly conductive. For the case of the predicted thermal damage shown in Figure 4b,c, it can be seen that the entire tumor area, plus margins of up to 1 mm around it, was completely destroyed ($\Omega = 100\%$). A comparison of temporal response curves for temperatures (Figure 4d) at different control points (Figure 3) within the tumor (P1)and at the tumor-gland boundaries (P2–P4) revealed that the temperature rise as well as the final value was higher at (P1) relative to the boundaries: P2 (top), P3 (bottom) and P4 (side). This phenomenon can be attributed to factors such as relatively low blood perfusion and high metabolic heat of the tumor leading to high retention of heat within the tumor [37]. However, at all the locations, the temperature plateaued after about 2–3 min. The consequence of the high temperature within the tumor is revealed in corresponding

predicted temporal curves for the thermal damage (Figure 4e), which shows that 100% thermal damage occurs faster in the innermost part of tumor (P1)—≈3 min—compared to the peripherals, which take up to about ≈10 min (P4). Consistent with the literature [33,38], the model predictions showed the dependence of thermal damage spatial profile on the temperature distribution, which decreased with distance away from center of the tumor (see Figure 4f).

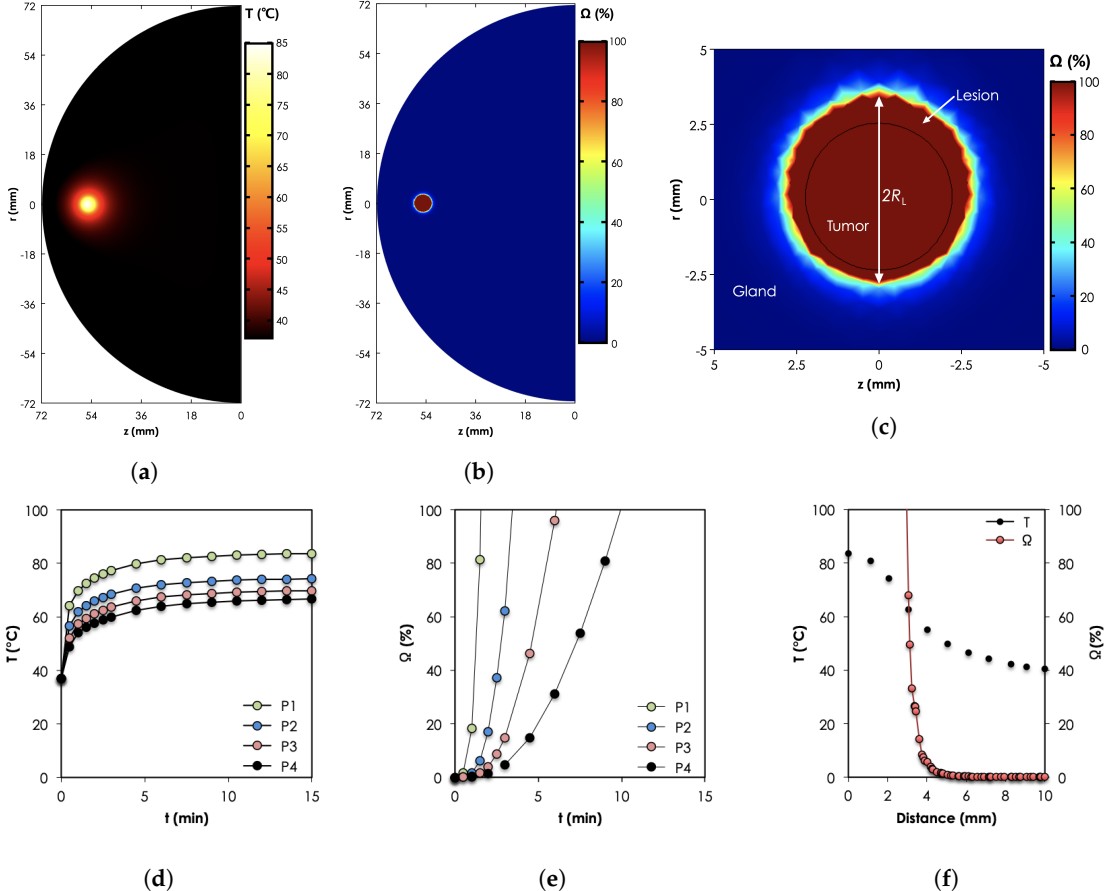

**Figure 4.** Simulation results. Cross-sectional view of the (**a**) temperature distribution, (**b**) thermal damage, (**c**) thermal damage showing the lesion parameter. Temporal response curves for (**d**) temperature and (**e**) thermal damage at the control points (P1–P4, cf. Figure 3). (**f**) Temperature and thermal damage as a function distance from P1. Simulation settings: $P_0 = 1$ W, $t = 15$ min and $T_{max} = 85\,^{\circ}$C.

Ablative temperatures between 60 and 100 $^{\circ}$C cause irreversible damages to key cytosolic and mitochrondrial enzymes [39,40]. For any tumor ablation therapy to be considered successful and thus reduce the chance of recurrence, it is critical to ensure that the entire volume of the tumor reaches therapeutic temperatures that ensures complete thermal damage ($\Omega = 100\%$). Such a goal can be achieved through the use of an appropriate maximum temperature, which takes into consideration the tumor dimensions. For NP assisted photothermal therapies such as the one being proposed in this study, maximum ablative tumor temperatures, $T_{max}$, can be controlled by varying parameters such as NP number density, $N$ (or volume fraction, $\phi_v$), the laser power, and treatment duration, $t$. To demonstrate this, a parametric study was used to determine $N$ required to achieve a given $T_{max}$ (70, 85, 100 $^{\circ}$C) and the corresponding volume of the lesion $V_L$ for different tumor sizes, $R$ (1, 2.5, 5 mm). $V_L$, was assumed to be spherical [41,42]; its radius, $R_L$, was calculated as half the axial length of the predicted cross-sectional area where $\Omega = 100\%$ (see Figure 4c). A summary of the results is presented in Table 1. The simulations were run with $P_0 = 1$ W and $t = 15$ min. Generally, it can be observed that $T_{max}$ required to achieve complete thermal damage increased with size of the tumor. For instance, $T_{max} = 70\,^{\circ}$C produced a lesion with $V_L = 2.95$ mm$^3$, which was insufficient

to completely ablate the entire volume of tumor with $R_T = 1$ mm ($V_L = 4.19$ mm$^3$). On the other hand, $T_{max} = 85$ produced a lesion with $V_L = 2.95$ mm$^3$, which was big ensure to ensure complete thermal damage. Since $P_0$ was held constant for all simulation, it meant that $N$ had to be increased to achieve the given $T_{max}$. The results reveal that $N$ required to achieve $T_{max} = 70$ °C decreased with tumor size. For instance, $N$ required to achieve $T_{max} = 70$ °C decreased from $112.37 \times 10^{14}$ mL$^{-1}$ to $5.54 \times 10^{14}$ mL$^{-1}$ when $R_T$ was increased from 1 to 5 mm. Lastly, the nanoparticle concentrations that were required to achieve the different values of $T_{max}$ corresponded to volume fractions in the range between 0.004% and 10.6%. A review of the nanoparticle delivery to tumors in the literature between 2006 and 2016 by Wilhelm et al. [43] revealed that only approximately 1% of administered nanoparticle dose reached the tumor. Therefore, it is important that the $\phi_v$ is kept at the low value for practical applications. This can be achieved by through several means such as increasing the laser power or exploiting the capability of the Fe$_3$O$_4$ NPs to generate synergistic heat during simultaneous exposure to NIR laser and alternating magnetic field as previously reported elsewhere [19].

**Table 1.** Comparison of volume, $V_L$, of predicted lesions and the number density, $N$, of nanoparticles (or volume fraction, $\phi_v$) used to achieve maximum tumor temperatures, $T_{max}$ (70, 85, 100 °C) in different tumor sizes, $R$ (1, 2.5, 5 mm). $R_L$ is the radius of the lesion.

| $T_{max}$ (°C) | $R = 1$ mm | | $R = 2.5$ mm | | $R = 5$ mm | |
|---|---|---|---|---|---|---|
| | $N(\phi_v)$ ($\times 10^{14}$/mL) | $V_L(R_L)$ (mm$^3$) | $N(\phi_v)$ ($\times 10^{14}$/mL) | $V_L(R_L)$ (mm$^3$) | $N((\phi_v))$ ($\times 10^{14}$/mL) | $V_L(R_L)$ (mm$^3$) |
| 70 | 112.37 (5.68%) | 2.95 (0.89) | 19.36 (0.06%) | 20.94 (1.71) | 5.54 (0.002%) | 44.00 (2.19) |
| 85 | 166.11 (8.17%) | 15.30 (1.54) | 27.60 (0.09%) | 128.45 (3.13) | 7.18 (0.003%) | 347.17 (4.36) |
| 100 | 221.34 (10.6%) | 24.43 (1.80) | 36.72 (0.13%) | 256.20 (3.94) | 9.26 (0.004%) | 998.31 (6.20) |

These predictions are consistent with previously reported experimental and computational results in the literature. Kannadorai et al. [44], developed a treatment planning model for the optimization to parameters such as laser power density, nanoparticle concentration and exposure time in an effort aimed at potential enhancement of treatment outcome. Their predictions showed that any change made to any of the parameters can be compensated by altering the remaining parameters. Using an integrated strategy that combined x-ray computed tomography or ex-vivo with a 4-dimensional FEM model, Maltzahn and co-workers [20] simulated photothermal heating with polyethylene glycol PEGylated gold nanorods (PEG-NR) and used the results to guided pilot therapeutic studies on human xenograft tumors in mice. Their simulations revealed the extension of thermal flux vectors from the region where PEG-NRs were located as well as the expected thermal profile.

## 3. Discussion

Generally, the efficacy and safety of NP-mediated PPTT depend on several independent factors such as the properties of nanomaterial (e.g., morphology, size distribution, optical absorption coefficient), biological identity (e.g., in-vivo circulation time, stability, tumor-homing) and irradiation protocols (e.g., laser beam power, shape, duration, cross-section, direction). Therefore, it requires an integrated strategy that combines experiments and models to optimally select and customize these parameters towards the realization of a reliable and efficient treatment outcome. Clearly, we acknowledge that the strategy we describe here is not exhaustive; however, our intention was to emphasize the need for a structured procedure that allows a quantitative assessment of the heat generation capabilities and predict critical optical properties of the nanoparticles that can be used in computational modeling.

We show that Fe$_3$O$_4$ NPs exhibit photothermal effects when irradiated with NIR (810 nm) light leading to photothermal generation, which increases with NP concentration and laser power. On the basis of the optical (Figure 1c) and structural (Figure 1b) properties, the absorption coefficient that

was used in the computational model was predicted with the Mie scattering theory. It is worth noting here that we used the Mie theory because the NPs were spherical [45], however, the photothermal effect is not unique to only spherical iron-oxide NPs but also cubic [19], hexagonal and wire-like [18]. For such non-spherical geometries, discrete dipole approximation—a discrete solution method of the integral form of Maxwell's equations, should be used [46]. Qin et al. [47] used a combination of the two methods to perform quantitative comparison of photothermal heat generation between gold nanospheres and nanorods. Estimation of $\eta_{exp}$, which describes how the NPs dispose (scattering plus absorption) the incident electromagnetic energy, has implications for NP concentration and laser beam power to be used. Although, it was beyond the scope of this work because it has been extensively studied previously [20,48], the biodistribution and effective tumor-homing following intratumoral or i.v. administration is key to the efficacy of treatment. To this end, techniques such as PEGylation and ligand-conjugation of the NPs have been shown to enhance and modulate their performance for biomedical applications and, thus, must be considered as part of efforts to fully characterize the nanoparticles for in-vivo applications.

Due to the complexities of multi-tissue breast tissue and different characteristics of tumors (size, location, shape), coupling of experimental measurements with computational modeling allows for the progressive selection, optimization and customization of parameters including NP concentration, irradiation protocols and treatment duration for in-vivo applications. This approach is essential for mitigation or prevention of collateral damage to healthy tissue surrounding the tumor. Here, we used optical absorption coefficient obtained via Mie theory predictions to develop a FEM model and used a temperature-controlled parametric study to demonstrate that the temperatures of different sized fibroadenomas can reach ablative levels leading to complete thermal damage ($\Omega = 100\%$) during irradiation with different laser powers. Several investigators have shown that the accuracy of FEM models for thermotherapy can be enhanced by using realistic geometries and material properties [20,21,34,44]. Although our model accounted for temperature dependence and blood perfusion effects, the multi layer geometry based on BIRADS [31] is generic and the distribution of the NPs was an assumption. Such simplification can have an adverse effect on integrity of the predicted values. Several reports have shown that using geometries that correlate with real anatomic datasets and include biodistribution data [20,34] have the potential to improve the accuracy of predictions. Elsewhere, such datasets have been obtained via noninvasive techniques such as X-ray computed tomography, sonography and ex-vivo spectrometry [20,34].

Finally, we acknowledge that Au NPs have been the prime candidates for photothermal therapy, however, it still remains an experimental cancer treatment due to issues related to their bio-persistent, which makes them potentially toxic and the use of high irradiation doses to achieve therapeutic temperatures due to the turbidity of biological tissues [28,49]. These issue have led to the recent interest in the photothermal properties of $Fe_3O_4$ NPs, which have been approved by the food and drugs administration (FDA). Furthermore, recent studies that have explored the simultaneously application (DUAL-mode) of both NIR laser and alternating magnetic field (AMF) to the $Fe_3O_4$ NPs have shown promising and interesting results. The studies found that the amount of heat generated with DUAL-mode equaled the sum of the heating for NIR laser or AMF only [19,26]. The essence of these results is that the use of the DUAL-mode can be used to overcome the challenges associated with the individual techniques.

## 4. Materials and Methods

### 4.1. Materials

The following materials were used in this study: $Fe_3O_4$ (99.5%, 15–20 nm) NPs (US Research Nanomaterials Inc., Houston, TX, USA).

### 4.2. Experiments

#### 4.2.1. MNP Characterization

Fe$_3$O$_4$ NPs were characterized by TEM (Philips CM10, Philips Electron Optics, Eindhoven, The Netherlands) and XRD (D8 FOCUS X-ray, Bruker AXS GmbH, Karlsruhe, Germany) for crystal structure and morphology and then UV-vis-NIR spectroscopy (GENESYS 10S UV-vis, Thermo Fisher Scientific, Madison WI, USA) in the wavelength range of 400–900 nm for absorption spectra.

#### 4.2.2. Photothermal Measurement in Water

The sample (Fe$_3$O$_4$ NPs in 0.5 mL of deionized water) contained in a 1.5 mL Eppendorf tube was irradiated by a NIR continuous laser at 810 nm (Photon Soft Tissue Diode Laser, Zolar Technology & MFG, Canada) with an external adjustable power, $P_0$ (0–3 W). The distance between the sample and the laser was 1–2 cm and the laser spot size was about 1 mm. The laser powers that were used was 0.5 and 1.0 W. Each sample was identically irradiated for 5 min. The resulting temperature rise was recorded by thermocouples (J-type, National Instrument, Austin, TX, USA) connected to a portable data acquisition system (NI USB-9222A, National Instruments, Austin, TX, USA) and recorded every 30 s with NI-DAQmx (National Instruments, Austin, TX, USA) and software (LabVIEW 8.6, National Instruments, Austin, TX, USA). All measurements were obtained in triplicate except stated otherwise.

The experimental photothermal conversion efficiency ($\eta_{\text{exp}}$) of the NPs was calculated directly from steady-state temperature increase as follows:

$$\eta_{\text{exp}} = \frac{Q_{\text{exp}}}{P(1 - 10^{-A_{exp}})} \tag{1}$$

where $Q_{\text{exp}}$ (W) was calculated with previously reported expression [47]: $16.855\Delta T$ (mW), $\Delta T$ is the temperature change, $P$ is the incident laser power and $A_{\text{exp}}$ is the absorbance of the Fe$_3$O$_4$ nanoparticles at 810 nm.

### 4.3. Models

#### 4.3.1. Optical Properties Predictions

The experimentally measured absorbance, $A_{\text{exp}}$, of a colloidal solution can be expressed in terms of predicted extinction cross section, $\sigma_{\text{ext}}$ as:

$$A_{\text{pred}} = N \frac{\sigma_{\text{ext}}}{2.303} d_0 \tag{2}$$

where $N$ (m$^{-3}$) is the number density of the NPs, $d_0$ (cm) is the path length of the spectrometer. For spherical, homogeneous and isotropic NPs, the "Mie scattering theory" [45,50] can be used to compute the exact values of the $Q_{\text{ext}}$, absorption ($Q_{\text{abs}}$) and scattering ($Q_{\text{sca}}$) efficiency as well as the anisotropy factor ($g$) as follows [46]:

$$Q_{\text{ext}} = \frac{2}{k^2} \sum_{n=1}^{\infty} (2n+1) \text{Re}(a_n + b_n) \tag{3a}$$

$$Q_{\text{sca}} = \frac{2}{k^2} \sum_{n=1}^{\infty} (2n+1) [(|a_n|^2 + |b_n|^2)] \tag{3b}$$

$$Q_{\text{abs}} = Q_{\text{ext}} - Q_{\text{sca}} \tag{3c}$$

$$g = \frac{4}{k^2 Q_{\text{sca}}} \sum_{n=1}^{\infty} \left[ \frac{n(n+2)}{n+1} \text{Re}(a_n a_{n+1}^* + b_n b_{n+1}^*) + \frac{2n+1}{n(n+1)} \text{Re}(a_n b_n^*) \right] \tag{3d}$$

where $k$ is the NP size parameter ($= 2\pi a/\lambda$). $a_n$ and $b_n$, the scattering coefficients in terms of the spherical Ricatti-Bessel functions, $\psi_n$ and $\eta_n$, respectively, are defined as:

$$a_n = \frac{\psi'_n(mx)\psi_n(x) - m\psi_n(mx)\psi'_n(x)}{\psi'_n(mx)\eta_n(x) - m\psi_n(mx)\eta'_n(x)} \tag{4a}$$

$$b_n = \frac{m\psi'_n(mx)\psi_n(x) - \psi_n(mx)\psi'_n(x)}{m\psi'_n(mx)\eta_n(x) - \psi_n(mx)\eta'_n(x)} \tag{4b}$$

where $m$ is the ratio of complex refractive index ($n_S = \sqrt{\epsilon_S}$) of the sphere to that of the surrounding medium ($n_m$) asterix ($*$) and prime ($\prime$) indicate complex conjugate and derivative with respect to $x$ and $mx$, respectively. The numerical calculations were performed with a python code implementation of the original algorithm published by Wiscombe [51]. The wavelength dependent complex refractive index, $n(\lambda)$, was obtained from Ref. [52].

To account for polydispersity, the size range of the nanoparticle was discretized into a varying number of terms ($n_t$) and then number-averaged to obtain the ensemble optical properties,

$$\overline{\sigma_k} = \frac{1}{n_t} \sum_{r=R_l}^{R_u} \sigma_k(r + i) \quad k = \text{ext, abs, sca}, \quad n_t = 2, 3, 4, \ldots, N \tag{5}$$

where $R_u$ and $R_l$ are the upper and lower limits of the NP size range, respectively. $i$ is the step size which is calculated as: $i = R_u - R_l/(n - 1)$ and $\overline{\sigma_k}$ is the mean $k$ (i.e., extinction, absorption, scattering) cross sections of the NP.

### 4.3.2. In-Vivo Predictions

The computational model is a multiphysics FEM model, thus, it took into account optical and thermal effects. Light distribution was based on the diffusion approximation of the transport theory [29] and temperature distribution by Pennes bio-heat transfer equation [30], which takes into account the effect of cell death on blood perfusion and the dependence of cell death and properties of the tissue. Cell death was determined by an Arrhenius based integral injury model [53].

***Light Distribution.*** The optical diffusion approximation of the transport theory [29] was used to describe light distribution due to the dominance of scattering over absorption in biological tissues. It is defined by:

$$\frac{1}{c_n}\frac{\partial}{\partial t}\varphi = D\nabla^2\varphi - \mu_a\varphi + S \tag{6}$$

$c_n$ (m s$^{-1}$) is the speed of light in a medium, $\varphi$ (W m$^{-2}$) is the fluence rate, $\mu_a$ (m$^{-1}$) is the absorption coefficient, $S$ (W m$^{-3}$) is the light source term and $D = \mu_a/\mu_{\text{eff}}^2$ (m) is the diffusion coefficient. $\mu_{\text{eff}} = \sqrt{3\mu_a(\mu_a + \mu'_s)}$ (m$^{-1}$) is the effective attenuation coefficient and $\mu'_s$ (m$^{-1}$) is the reduced scattering coefficient. Assuming that the light source was a continuous wave Gaussian NIR laser beam that was incident onto the breast model, the $\varphi$ can be defined by

$$\phi(\vec{r}) = \frac{P_0 \exp(-\mu_{\text{eff}}\vec{r} \cdot \hat{n})}{4\pi D r} \tag{7}$$

where $P_0$ is the laser power and $\hat{n}$ is the direction of the beam. A summary of the values of the optical properties of the tissue used in the simulation is presented in Table 2.

**Table 2.** Optical properties of the biological domains that were used in the simulations. The values were obtained from Refs. [34–36].

| Tissue | Coefficients, (m$^{-1}$) | | Refractive Index, (1) |
|---|---|---|---|
| | Absorption, $\mu_a$ | Reduced Scattering, $\mu'_s$ | $n$ |
| Fat [35] | 3 | 950 | 1.455 |
| Gland [35] | 6 | 1100 | 1.4 |
| Muscle [34] | 23 | 130 | 1.37 |
| Tumor [36] | 7 | 1400 | 1.37 |

*Temperature Distribution.* The Pennes bio-heat transfer equation [30] was used to estimate the temperature distribution. An additional term was added to account for the external heat source. The resulting equation is given by:

$$\rho c_{\mathrm{p}} \frac{\partial T}{\partial t} = \lambda(T)\nabla^2 T + \rho_{\mathrm{b}} c_{\mathrm{b}} \omega_{\mathrm{b}}(\Omega)(T_{\mathrm{b}} - T) + Q_{\mathrm{met}} + Q \tag{8}$$

where $\rho$ (kg m$^{-3}$) is the density, $c_{\mathrm{p}}$ (J kg$^{-1}$ K$^{-1}$) is the specific heat capacity at constant pressure. $\lambda(T)$ (W m$^{-1}$ K$^{-1}$) is the temperature dependent thermal conductivity, which is assumed to be a linear function defined by [54]:

$$\lambda(T) = \lambda_{(37\,^{\circ}\mathrm{C})}[1 + 0.0028(T - 293.15K)] \tag{9}$$

where $T$ (K) and $T_{\mathrm{b}}$ (K) are the normal body and arbitrary temperatures, respectively. $\rho_{\mathrm{b}}$ is the density of blood, $c_{\mathrm{b}}$, the specific heat capacity of blood and $\omega_{\mathrm{b}}(\Omega)$ is the coefficient of blood perfusion assumed to be dependent on the cell damage, $\Omega$, and defined by [33,38]:

$$\omega_b(\Omega) = \begin{cases} \omega_b^0 & \text{if } \Omega = 0 \\ (1 + 25\Omega - 260\Omega^2)\omega_b^0, & \text{if } 0 < \Omega \leq 0.1 \\ (1 - \Omega)\omega_b^0, & \text{if } 0.1 < \Omega \leq 1 \\ 0, & \text{if } \Omega > 1 \end{cases} \tag{10}$$

$\omega_b^0$ (s$^{-1}$) is the baseline coefficient of blood perfusion. $Q_{\mathrm{met}}$ (W m$^{-3}$) is the metabolic heat. $Q$ accounts for external heat sources, which varies for the different domains of geometry. The heat generated after the absorption of NIR light is defined as $\mu_a \varphi(r)$ (W m$^{-3}$) and $N\sigma_a \varphi(r)$ (W m$^{-3}$) for the tissue and tumor domains respectively. $N$ is the number volume of Fe$_3$O$_4$ NPs and $\sigma_a$ (m$^2$) is the absorption cross-section of nanoparticles. Table 3 presents a summary of the values of the thermo-physical properties that were used in the simulation.

**Table 3.** Thermo-physical properties of the biological domains that were used in the simulation. The values were obtained from Refs. [31,33]

| Tissue | Specific Capacity Heat $c$ [J (kg K)$^{-1}$] | Thermal Conductivity $\lambda$ [W (mK)$^{-1}$] | Density $\rho$ [kg m$^{-3}$] | Metabolic Heat $Q_{\mathrm{met}}$ [W m$^{-3}$] | Blood Perfusion $\omega_{\mathrm{b}}$ [s$^{-1}$] |
|---|---|---|---|---|---|
| Fat | 2348 | 0.21 | 911 | 400 | 0.0002 |
| Gland | 2960 | 0.48 | 1041 | 700 | 0.0005 |
| Muscle | 3421 | 0.48 | 1090 | 700 | 0.0008 |
| Tumor | 3770 | 0.48 | 1050 | 8720 | 0.0001 |
| Blood | 3617 | - | 1050 | - | - |

*Thermal Damage.* The Arrhenius injury model was used to estimate tissue destruction. The model, which relates temporal temperature to cell death, is defined by [53]:

$$\Omega(\tau) = A \int_0^\tau \exp\left(\frac{-E_a}{RT(t)}\right) \mathrm{d}t \tag{11}$$

where $E_a$ (J mol$^{-1}$) is the activation energy, $A$ (s$^{-1}$) is a scaling factor and $R = 8.3$ (J mol$^{-1}$K$^{-1}$) is the gas constant. The values for $E_a$ and $A$ were obtained from Ref. [33] as 302 kJ mol$^{-1}$ and $1.18 \times 10^{44}$ s$^{-1}$ respectively. $\Omega = 1$ corresponds to the 100% irreversible cell damage.

*Model Implementation.* This FEM model was developed with the COMSOL Multiphysics 5.2 software package (Comsol, Inc. Burlington MA, USA). All properties and dimensions were added explicitly to the FEM model as parameters and variables under the "Global Definition" and "Model" nodes, respectively. Equations (9) and (10) were added as analytic functions under the "Global Definition" node. The 2D axisymmetrical model was used to reduce simulation time.

The light distribution was achieved by implementing Equation (5) as an analytic function. The temperature distribution was achieved using the bio-heat heat transfer application mode. Each tissue was represented with a separate "biological tissue" node. The boundary and initial conditions were specified as follows: a Dirichlet condition, T = 37 °C, at $\Gamma_1$; a Neumann condition, $\mathbf{n} \cdot (\lambda \nabla T) = h \cdot (T_{\text{ext}} - T)$ at $\Gamma_2$ where the heat transfer coefficient, $h$ was equal to 13.5 Wm$^{-2}$K$^{-1}$ and $T_{\text{ext}} = 25$ °C and continuity, $\mathbf{n} \cdot (\lambda_1 \nabla T_1 - \lambda_2 \nabla T_2) = 0$ at all interior boundaries. A temperature of 37 °C (for the normal body) was used as the initial temperatures in all domains of the model. The heat source was added to the bio-heat transfer application mode as a user-defined heat source.

The cell death model was implemented with the "Coefficient Form PDE" application mode. To achieve a time integration, the coefficients $d_a$ and $f$ were set to 1 and Equation (11), respectively. All other coefficients were set to zero. The initial conditions were: $S = \partial S / \partial t = 0$.

In order to enhance the accuracy of results, we resolved the model with successively smaller element sizes and compared results, until an asymptotic behavior of the solution emerged. The comparison was done by analyzing the temperature at the interface between tumor and the tissue. The choice of 2D-axisymmetric model allowed for the use of a physics-controlled mesh with the triangular element with sizes: maximum = 0.24 cm and minimum = 0.0024 cm for the tumor region and element sizes: maximum = 0.42 cm and minimum = 0.018 cm for the other regions. This resulted in 2656 domain elements and 277 boundary elements. The numerical solutions were obtained using the time-dependent solver "GMRES" with its default settings. The simulations were run on a mid-range workstation with Intel(R) Xeon(R) E5-1620 CPU and 8 GB of RAM.

## 5. Conclusions

Recent developments in imaging techniques have led to early detection of small fibroadenomas. Although observation is recommended for such cases, the agitations by some women due to fear of malignancy [12] coupled with recent report of 41% increase in cancer risk for women diagnosed with fibroadenomas [13] justify the need to develop techniques that can destroy these tumors with minimal or no side effects. We believe our findings demonstrate the potential of NP-mediated photothermal therapy for destroying fibroadenomas. However, we acknowledge the limitations of the study and understand that a future study should incorporate the important aspects discussed earlier so that a proper assessment can be made. Our long term goal is to develop a non-aggressive and noninvasive treatment method for such benign tumors, which is becoming a growing public health concern.

**Author Contributions:** Conceptualization, A.Y. and K.K.-D.; methodology, I.B.Y., S.W.K.H., Y.K.K.-A., A.Y. and K.K.D.; software, A.Y. and K.K.-D.; formal analysis, I.B.Y., S.W.K.H., Y.K.K.-A., A.Y. and K.K.D.; investigation, I.B.Y., S.W.K.H., Y.K.K.-A., A.Y. and K.K.D.; writing—original draft preparation, A.Y. and K.K.-D; writing—review and editing, I.B.Y., S.W.K.H., Y.K.K.-A., A.Y. and K.K.D.; project administration, K.K.-D.; funding acquisition, K.K.-D. All authors have read and agreed to the published version of the manuscript.

**Funding:** This research was funded by the Building a New Generation of Academics in Africa (BANGA-Africa) project, which is funded by the Carnegie Corporation of New York.

**Conflicts of Interest:** The authors declare no conflict of interest.

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
