# Peer review of "Destruction of Fibroadenomas Using Photothermal Heating of Fe3O4 Nanoparticles: Experiments and Models"

_applsci, doi:10.3390/app10175844_

Round 1

Reviewer 1 Report

The paper reports theoretical and experimental study of Fe3O4 nanoparticles for the treatment of breast lesions via photothermal therapy. The authors have demonstrated the optical and thermal characterization of Fe3O4 nanoparticles and modeling of the effect of the nanoparticle for heat mediated damage of tumor in the breast. The following comments should be addressed before the paper can be considered for publication.

  • It is not clear why the authors are claiming Fe3O4 nanoparticles to be plasmonic. Noble metals and highly doped semiconductors show plasmonic properties. Mie scattering of dielectric particle is well known but that does not necessarily mean that they are plasmonic. The authors should correct this concept in the paper or provide suitable backing for their claim.
  • Y axis labels are missing in figure 1. Also, the quality of figure 1 should be improved significantly – figure texts are unreadable.
  • The literature from which the authors have used the tissue properties should be cited in the main text (page 5).
  • Simulation with other dielectric particle like silicon dioxide can be demonstrated as a control to highlight advantage of Fe3O4
  • Use of gold nanoparticles is a well established technique for photothermal therapy. Gold is also known to be biocompatible. The authors should explain more clearly how Fe3O4 nanoparticles better in this regard.

Reviewer 2 Report

The authors have proposed the use of ferric oxide nanoparticles for the plasmonic photothermal therapy of fibroadenomas using FEM. I have some reservations before recommending this manuscript for publication. Please clarify the points below:

  1. Can the authors compare the more conventionally used gold nanoparticles with iron oxide particles in terms of the photothermal damage caused by them? A comparison should be put in the discussion section.
  2. Fig. 1 texts in the panels are not visible. Please adjust the font size.
  3. Page 3, line 105: approach saturation at 5 min. Fig. 2a and 2b do not look like saturating at 5 min. Please provide extended temporal data.
  4. Please define the points P1-P4 used in the description of figure 3, before using them. Consider showing figure 4 before figure 3.
  5. Right now the connection between the experiments and simulations are not obvious. Please describe how the experimental measurements performed in Fig. 1 and 2 helped in coming up with the modeling. How does no. of particles, N used in the FEM compare to the concentration of particles used in the experiments. 
  6. Can the authors mention the volume fraction of the nanoparticles inside the tumor required to achieve the reported thermal damage. It is important to ensure that this number is practically achievable.
  7. The modeling predictions described in page 6, line 160 onwards, need to be re written. They are difficult to follow and not interesting. 
  8. Page 7, expand BIRADS
  9. Page 7, line 220: provide references 

Overall, I feel that the lack of experimental validation, of the localized photothermal damage claimed in this manuscript, makes the study weak. The authors might consider a demonstration of the localized heating in a tissue phantom model. 

Reviewer 3 Report

After reading the paper "Destruction of Fibroadenomas Using Plasmonic Photothermal Heating: Experiments and Models", I find this latter not suitable for publication in Applied Sciences due to the fact Fe3O4 nanoparticles in visible and near-infrared ranges of spectrum do not have plasmonic resonances. Thus, here in this paper, they cannot say that it is a plasmonic effect. The authors must change their explanation and more generally their paper.

Round 2

Reviewer 1 Report

Authors have addressed the comments sufficiently. 

Reviewer 2 Report

The authors have answered all the concerns. I recommend publication

Reviewer 3 Report

After reading the revised version of this manuscript and that authors have corrected their grave error concerning the "plasmonic" aspect of their paper, this latter can be accepted for publication.